# Personalized Medicine Based on the Pathogenesis and Risk Assessment of Endodontic–Periodontal Lesions

**DOI:** 10.3390/jpm12101688

**Published:** 2022-10-10

**Authors:** Keiso Takahashi, Kousaku Yamazaki, Mikiko Yamazaki, Yasumasa Kato, Yuh Baba

**Affiliations:** 1Division of Periodontics, Department of Conservative Dentistry, Ohu University School of Dentistry, 31-1, Misumido, Tomita-machi, Koriyama City 963-8611, Fukushima, Japan; 2Division of Oral Pathology, Department of Oral Medical Science, Ohu University School of Dentistry, 31-1, Misumido, Tomita-machi, Koriyama City 963-8611, Fukushima, Japan; 3Department of Oral Function and Molecular Biology, Ohu University School of Dentistry, 31-1, Misumido, Tomita-machi, Koriyama City 963-8611, Fukushima, Japan; 4Department of General Clinical Medicine, Ohu University School of Dentistry, 31-1, Misumido, Tomita-machi, Koriyama City 963-8611, Fukushima, Japan

**Keywords:** endodontic–periodontal lesions, multifactorial disease, pathogenesis, classification, risk assessment, root canal retreatment, personalized medicine

## Abstract

Endodontic–periodontal lesions (EPLs) are chronic inflammatory lesions in the mouth caused by multiple factors. Both periapical and marginal periodontitis are characterized by infection and inflammation around the affected teeth, suggesting that the theory of complex systems might describe the progression of EPL. The diagnosis and treatment of EPLs are complicated by variations of this condition and difficulties distinguishing EPLs from other diseases. Technological advances in diagnostic and treatment methods, including cone beam computed tomography, microscopy, mineral trioxide aggregates, and periodontal regenerative treatment, have improved outcomes, even in untreatable teeth. However, treating EPLs with iatrogenic problems and/or severe periodontitis remains challenging. Assessing the risk of each EPL based on the possible pathogenesis of each EPL is essential for determining individualized treatment and optimizing personalized medicine for individual patients.

## 1. Introduction

Diagnostic difficulties frequently arise in dental practice. These difficulties can make the optimization of dental treatments unclear, as uncertainties in diagnosis may increase the likelihood of treatment risks and iatrogenic errors. For example, clinical findings in patients with endodontic–periodontal lesions (EPLs) can vary widely and include deep periodontal pockets, bleeding on probing, suppuration, and bone resorption [1,2,3] resulting from inflammation around and the destruction of affected teeth. EPLs might be accompanied by infection with multiple types of bacteria, and microflora in infected root canals and periodontal pockets have been reported to be similar [4,5].

Periodontitis is characterized by chronic/acute infection and inflammation of tooth-supporting tissues and by the loss of connective tissue and alveolar bone [6]. This disease is multifactorial, with patients and teeth differing in susceptibility and resistance to periodontitis depending on individual causal and risk factors [7]. Simple mathematical formulas cannot determine the progression of periodontitis; rather, it may be determined using the theory of complex systems [8]. Approximately 300,000 to 400,000 periodontal ligament fibers were shown to be present on the surface of each root, with the progression of periodontitis determined using a nonlinear chaotic model [9,10]. The pathogenesis of aggressive periodontitis has been determined by recursive division analysis and immunological assessment of neural networks [11]. The theory of complex systems may also explain the wide heterogeneity of clinical symptoms and the difficulty of treatment and prognosis in patients with EPLs. In addition, endodontic failure caused by iatrogenic errors can affect the progression of EPLs, making a differential diagnosis of EPLs before treatment more difficult than that of periodontitis alone. 

The treatment of EPLs should include the removal of bacterial biofilm and products of infectious agents, both in infected root canals and periodontal pockets. The prognosis of EPL has been reported to be poorer than endodontic lesions in patients without periodontitis [12]. EPL mainly involving periodontitis tends to show a poorer prognosis because regenerative periodontal therapy is essential for teeth that are beyond treatment. Moreover, treatment outcomes are dependent on several factors, including the skill of the dentist, patient compliance, healing ability, and the quality of the supportive periodontal therapy program. 

Patient-associated factors, including smoking habits, diabetes, parafunction, and socioeconomic status, are involved in the pathogenesis of periodontitis. Similarly, iatrogenic errors during endodontic treatment can influence the prognosis of affected teeth [13,14]. The re-evaluation of endodontic healing before nonsurgical and/or surgical periodontal therapy for EPL is therefore important [15]. Algorithms and decision trees for the diagnosis and treatment of EPL have been proposed [16,17]. 

Technological advances, including cone beam computed tomography (CBCT), microscopy, mineral trioxide aggregates (MTA), and periodontal regenerative treatment, have improved the diagnosis and treatment of EPL. In addition, pre-treatment risk assessment in patients with EPLs can determine the likelihood of a good prognosis. These technologies may improve precision medicine for patients with EPLs. 

The objectives of the present review are to highlight the potential causes and risk factors associated with each EPL and the uncertainties involved in differentiating EPLs from other diseases because of their multifactorial nature and similar clinical features. Moreover, this review proposes practical methods of determining treatment and the advantages of precise and personalized medicine for each EPL based on risk assessments in individual patients and an interdisciplinary approach.

## 2. Definition of EPLs

The relationship between pulpal and periodontal disease was first described in 1964 [2]. That study was the first to use the term “retrograde periodontitis”, which indicated periapical periodontitis or periradicular disease. EPLs could be differentiated from marginal periodontitis, so they were therefore defined as lesions due to varying degrees of inflammatory symptoms in both pulpal and periodontal tissues. The simultaneous occurrence of endodontic problems and periodontitis tends to complicate differential diagnoses. Although EPLs have been defined, this definition is not based on rigid scientific evidence or an exact mechanism. EPLs can be classified by etiologic factors, manifestations, and mechanisms of action, with these classifications likely to improve based on new evidence or consensus. 

## 3. Classifications of EPLs

A classification of EPLs is summarized in Table 1 [18,19,20,21,22,23,24,25,26,27]. EPLs were originally classified into five patterns based on primary lesions [18], but this classification was pragmatic rather than explanatory, and the exact pathogenesis of each remained unclear. An alternative classification based mainly on etiology included both the cause of disease and treatment needs [19]. This classification also included other problematic conditions, such as root perforations, root resorption, root fractures, and grooves, with clinical symptoms similar to those of EPL. An additional classification was based on treatment needs rather than on possible etiology or diagnostic processes [20]. 

The latest classification by Herrera et al. includes the presence or absence of root damage and sub-classifies into three grades [27]. However, this classification is insufficient for the personalized diagnosis of EPLs and we need to differentiate EPLs from other diseases, including vertical root fractures, iatrogenic perforations, internal/external root resorption, and cemental tears that share similar clinical features with EPLs. We should establish the basic concept for diagnosing each EPL in patients in which possible individual risk factors are associated with each patient with periodontitis [28].

EPLs have also been classified into six categories based on the origin of periodontal pockets [21], although this classification was not used in later editions but was replaced by another type of classification [22]. In 1999, the World Workshop for the Classification of Periodontal Diseases classified periodontitis associated with endodontic disease into three categories [23], whereas another EPL classification system was based on criteria associated with communications between endodontic and periodontal lesions [24]. EPL has also been classified into four types: endodontic; periodontal; true-combined; and iatrogenic lesions, with the latter including root perforation, overfilling of root canals, coronal leakage, trauma, chemically induced root resorption, and vertical root fractures (VRF) [25]. Another new classification system for EPL was based on the primary disease with its secondary effects and included the concept of iatrogenic periodontal lesions [26]. 

In 2018, the World Workshop for the Classification of Periodontal and Peri-implant Diseases and Conditions first classified EPL into two categories according to the presence or absence of root damage by fractures and/or perforations, followed by classification according to the presence or absence of periodontitis that was or was not associated with trauma and iatrogenic factors [27]. EPL was further subdivided into three categories based on the degree of tissue damage. This classification system was primarily related to periodontic problems, with endodontic problems, including iatrogenic errors, not being effectively evaluated. Infected root canals are often damaged by poor endodontic treatment, affecting the outcome of endodontic retreatment [29,30]. Individualized assessment of risk for each EPL is crucial for both classification and determining treatment. 

Inadequate dental treatment, especially endodontic failure, could be a risk for EPLs as well as peri-implantitis [31]. The prevalence of endodontically treated teeth was higher in Japan than in Europe and the USA [32], and iatrogenic errors have been associated with refractory periapical periodontitis in general practices in Japan [33]. Both endodontic and periodontal treatment may injure treated teeth, emphasizing the need for the continuous training of dentists. 

## 4. Clinical Manifestations and Examinations

The most common clinical manifestations of EPL include abscess formation, deep periodontal pockets, and bone resorption on X-ray photographs and/or CBCT due to bacterial infection and host inflammatory reactions. Patients may experience discomfort, occlusal and/or spontaneous pain, purulent exudate, sinus tract infection, gingival swelling, and increased mobility of affected teeth. Incomplete or excessive root canal fillings and apical radiolucency on X-ray suggest that periapical periodontitis resulting from inadequate root canal treatment is usually involved in the pathogenesis of EPL. 

Patients suspected of EPL should be evaluated by endodontic and periodontal examinations, including electric pulp tests, pocket probing, X-ray photographs, CBCT, and medical interviews. Other factors that should be assessed include personal data, such as age, history of the disease, iatrogenic errors, trauma, and susceptibility to periodontitis. A full mouth examination is recommended to evaluate the severity of periodontitis and its risk for progression, and an occlusal examination is required to assess the possible causes of VRF, including parafunction, night bruxism, and tooth-contacting habits. Surgical inspection may reveal tooth cracks, VRF, perforation, and cemental tears, which can cause inflammatory symptoms. 

## 5. Risk Assessment of EPLs

Many factors have been associated with each EPL (Figure 1), including factors associated with periapical and marginal periodontitis. Dental caries, trauma, failure of endodontic treatment (iatrogenic factors), and advanced periodontitis, including retrograde pulpitis and furcation involvement, may affect periapical outcomes. Other factors include the clinical experience and skills of the dentist. To date, however, few diagnostic tools are available to determine endodontic failure due to various iatrogenic errors before treatment. 

Although the probability of success should be assessed before treatment, few methods are currently available [16,17]. Many factors have been associated with the failure of both endodontic and periodontal treatment methods, although these have not been quantified well, especially as the treatment skill of dentists and the healing ability of patients have been found to vary widely.

## 6. Differential Diagnosis of EPLs and Other Diseases

Several disease states, including root fracture, root perforation, internal/external root resorption, and cemental tears, share similar clinical features with EPLs. We need to diagnose them adequately before deciding on a treatment plan, although this is not always performed in some cases. Then, surgical inspection should be performed for uncertain lesions.

### 6.1. Root Fracture

Periodontal disease, caries, and root fracture of endodontically treated teeth with metal posts are major causes of teeth loss [34], with VRF being especially common in elderly patients. VRF may or may not be accompanied by bone resorption around affected teeth, with the incidence of VRF being higher in non-vital than in vital teeth. Deep periodontal pockets at restricted sites (narrow areas) suggest the presence of VRF, making the probing technique and clinician experience important. 

VRF occurring at the apex or middle of the root is not accompanied by a deep pocket in the early stages, although patients experience slight percussive pain and/or discomfort. Close observation and/or surgical inspection is required for a differential diagnosis. 

### 6.2. Root Perforation

Many endodontic failures may be caused by iatrogenic errors, such as perforations and the presence of broken instruments in curved canals of molar teeth. These errors have been attributed to poor endodontic treatment skills, making infection control difficult or impossible. 

Perforations can be categorized as iatrogenic (e.g., poor root canal preparation, post preparation, chemicals for breaching) or non-iatrogenic (e.g., caries, external and internal root resorption). Iatrogenic perforations can be further sub-classified as furcal, lateral, stripping, and apical foramen perforations, with perforation at the periapical foramen being the most common type of root damage caused by iatrogenic events [35,36]. Conceptually, suppressing the infection and adequate perforation repair should result in a good prognosis. Repeated root damage may also cause infectious reactions and damage around periapical tissues. 

Endodontic failure is the most common cause of iatrogenic disease of non-vital teeth. Root canal treatment is not always successful, indicating the need to preserve as much dental pulp as possible. Iatrogenic errors, such as ledges and root perforations, have been reported, and the frequency of errors was significantly greater in molars than in anterior teeth [37,38]. Success rates are the lowest in molars (26%), followed by premolars (49%) and anterior teeth (52%) because of the complex anatomy of the root canal system in molars and the difficulty of endodontic treatment by dentists unfamiliar with the proper procedures [39,40]. These findings indicate that current education for dental students is insufficient and that more training on endodontic treatment methods is required to reduce iatrogenic errors.

### 6.3. Internal and External Root Resorption

Root resorption in permanent teeth is usually pathological and can be classified by location (internal/external, cervical/apical) [41]. Internal resorption may induce pulp necrosis, root perforation, and bone resorption. However, periodontal surgery may not be required, with nonsurgical endodontic treatment using an operating microscope being sufficient in many patients. Endodontic treatment and root repair with MTA under microscopic examination have been associated with good patient prognosis. In contrast, external resorption can damage the cementum, root, and periodontal ligament, with surgical inspection and/or periodontal surgery being essential for diagnosis and treatment [42]. 

### 6.4. Cemental Tears

The pathogenesis of cemental tears (CTs) has not been studied as thoroughly as root resorption because CTs tend to be found accidentally during periodontal surgery and after tooth extraction. Aging, cementum thickening, and occlusal trauma may be involved in the etiology of CTs, although knowledge of CTs remains fragmented and based on a limited number of case reports [43,44,45]. Few animal or intervention studies to date have evaluated CTs. CTs tend to be found in vital and single-rooted teeth. The prevalence of CTs appears to be low, making them difficult to diagnose at early stages.

## 7. Treatment Modalities

Endodontic and periodontal treatment can be applied to root perforation and internal/external resorption with or without a microscope (Figure 2 and Figure 3). Cemental tears are often found accidentally during periodontal surgery (Figure 4). These are not EPLs; however, some clinical cases show similar clinical symptoms to EPLs, and they are sometimes difficult to differentially diagnose.

Charts have been proposed for determining periodontal treatment methods for both tooth retention and extraction [46]. These charts have assessed the possible risk of each factor. Similar charts may also be useful in determining treatment for EPLs.

The therapeutic strategy for EPLs usually consists of initial endodontic treatment, followed, if necessary, by additional periodontal treatment, with the latter being dependent on the outcomes of endodontic treatment of the affected teeth. Periodontal defects that communicate with periapical lesions and appear to be hopelessly diseased on X-ray radiographs or CBCT may have a favorable prognosis if they are of endodontic origin (Figure 5). EPLs that are conventionally defined as untreatable on X-ray radiographs or CBCT may not be beyond treatment, as the cementum and periodontal ligaments may be intact and not affected by periapical infection and inflammatory reactions. Conventional root canal treatment is the most common therapeutic choice for periapical infections and inflammation [47,48], although its success rates in damaged roots are relatively low [29,30].

Persistent periapical lesions may ultimately require endodontic surgery, such as apicoectomy and intentional replantation (Figure 6) [49]. Although this type of lesion is in the endodontic domain and is caused by iatrogenic errors, proper and successful treatment requires not only endodontic surgery but also concurrent guided tissue regeneration (GTR) [16,50]. 

EPLs with a completely compromised periodontium can be defined as beyond treatment and treated by both periapical surgery and periodontal regenerative treatment with GTR of the membrane and/or biological medications, such as enamel matrix derivative or fibroblast growth factor-2 (Figure 7 and Figure 8). A tooth with EPL accompanied by severe periodontitis can be diagnosed as a high-risk or an untreatable tooth (Figure 8), as it includes combined endodontic and periodontal lesions. Predictable and successful treatment of these truly combined lesions, in which periodontal ligaments and cementum are affected by an inflammatory reaction, remains challenging even for specialists. 

Adequate diagnosis and treatment by specialists, accompanied by good patient compliance and an excellent supportive periodontal therapy program, have resulted in a good prognosis in selected patients [51,52]. The management of individual predisposing and risk factors, such as smoking, parafunction, diabetes, malocclusion, and regular recall, is also essential for better outcomes.

In patients undergoing endodontic retreatment, access to apical infection may be compromised by iatrogenic errors in root canal preparation during previous treatment or by an inability to fully negotiate canal blockages due to root damage and natural or artificial materials. Treatment recovery following endodontic failure is often challenging (Figure 6 and Figure 7), with endodontic surgery under microscopy performed whenever possible. Even with the use of a microscope, however, the success rate is lower in patients with than without EPLs [49], indicating that the treatment of EPLs is difficult at present and suggesting the need for additional technology and special training of dentists on methods of regenerative therapy. 

Differential diagnosis distinguishing between EPLs and other disease conditions is not always possible before treatment. Diagnostic treatment, consisting mainly of surgical inspection, is required to resolve the diagnosis. VRF, root perforations, CTs, deep grooves such as the palate–gingival groove [53], furcation involvements, and iatrogenic events have been detected during and/or after treatment. Treatments based on the status of affected teeth, including tooth extraction, root repair, apicoectomy, and intentional replantation, are selected to optimize treatment outcomes. The choice to retain or not retain untreatable teeth is usually based on clinician judgment, with tooth extraction sometimes being the best choice for teeth with VRF.

## 8. Discussion

Despite the many classifications and treatment modalities of EPLs, treatment of these combined endodontic and non-endodontic problems remains challenging, with outcomes differing among different dentists. Endodontic retreatment is associated with a relatively high failure rate, indicating the need for surgical treatment combined with both endodontic and periodontal regenerative treatment of teeth that are beyond treatment (Figure 6 and Figure 7). That is, treatment outcomes depend on the status of the affected teeth diagnosed with EPLs. 

The uncertainty of diagnosis and dental treatment make decision-making difficult. Therefore, providing this information to patients may better enable them to select dentists suitable to treat their dental problems. 

Although collaboration between an endodontist and a periodontist is recommended [25], collaborations between general practitioners and specialists in Japan are generally poor, and remote guidance by specialists using the internet may be useful in the future. 

CBCT and microscopy are essential for precise endodontic diagnosis and treatment. Following endodontic treatment, patients should undergo additional treatment, such as surgery, including endodontic surgery, regenerative surgery, or surgical inspection. Additional clinical training is essential for both root canal preparation and periodontal surgery. Ozone and laser treatments of EPLs may show beneficial effects [54,55]. 

Although a significant amount of information is available for the diagnosis and treatment of EPLs, many questions remain unanswered [56]. EPL may be treatable if lesions are diagnosed at an early stage and treatment plans are determined quickly. Lesions may not be treatable if they include severe periodontitis around the affected teeth and VRT. 

Success rates tend to be lower after endodontic retreatment than after initial treatment, suggesting that biofilm infection may continue and affect the healing of EPLs, although endodontic retreatment is often effective even in patients with advanced EPLs [57] (Figure 5). Endodontic and periodontal education and practical training of dental students, including specialists, may improve the treatment outcomes. Patient education is also important in preventing the progression of oral diseases, including caries and periodontal disease.

Personalized medicine aims to individualize and optimize treatment and care, based on each person’s unique characteristics, including genetic, environmental, and clinical profiles [58]. Evidence-based medicine is based on a similar idea of optimizing individualized medicine by integrating each clinician’s expertise with the best scientific evidence available [59]. Clinical experience, diagnostic ability, and excellent surgical skills are crucial for the treatment of EPLs, especially in teeth deemed beyond treatment (Figure 7 and Figure 8). The primary goal of EPL treatment must be to remove the infection. In addition, optimal outcomes require the management of causal/risk factors in individual patients (Figure 9).

The definition of EPLs is still controversial, as described in Table 1, because the pathogenesis of EPLs has not been thoroughly investigated, and we could not obtain enough information for the diagnosis of EPLs. Further training for endodontic retreatment and regenerative periodontal treatment is required for the treatment of EPLs that are considered high risk or untreatable (Figure 7 and Figure 8). In addition, we need to obtain a basic concept of individualized risk assessment for each EPL, as described in Figure 1 and Figure 9, to create personalized medicine.

The ability of patients to heal also varies, and systemic conditions, such as diabetes mellitus, may reduce their ability to heal and worsen endodontic treatment outcomes [60,61]. Similarly, hyperlipidemia and obesity may affect peri-implantitis and periodontitis, respectively [62,63]. Therefore, further basic and clinical research are required to assess the associations of treatment outcomes with the healing ability of patients. 

## 9. Conclusions

EPLs are chronic inflammatory lesions caused by multiple factors, and treatment outcomes depend on the pathogenesis and progressive stage of each lesion. Differential diagnosis of EPLs and other disease conditions, including VRF, root perforations, and cemental tears, is sometimes difficult, regardless of the use of a microscope and a CBCT examination. Basic education for dental students about differential diagnosis and endodontic and periodontal treatments may be insufficient. Therefore, additional training is required to develop those dentists familiar with endodontic and periodontal diagnosis and treatments based on individualized risk assessments.

## Figures and Tables

**Figure 1 jpm-12-01688-f001:**
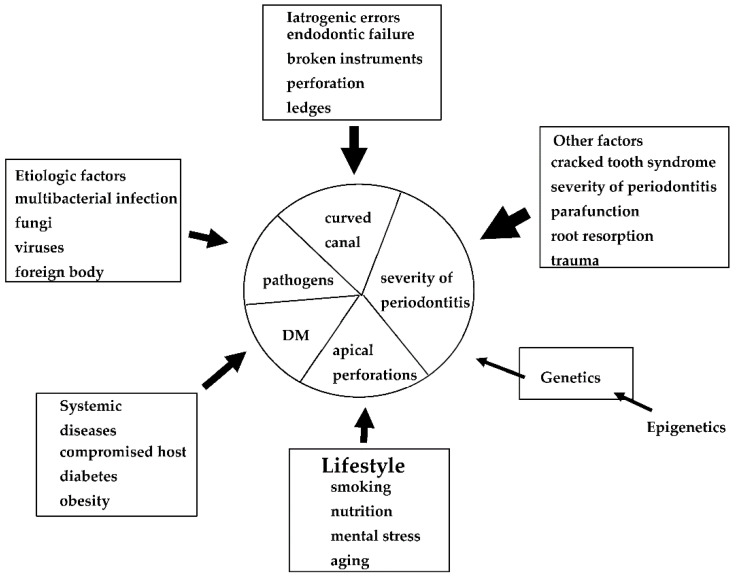
Possible causes, risk factors, and predisposing factors associated with EPLs. These factors may influence the progression and pathogenesis of each EPL.

**Figure 2 jpm-12-01688-f002:**
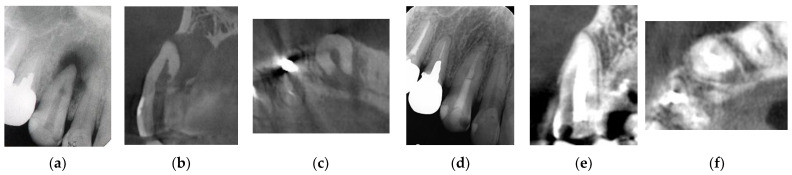
Clinical case 1. Male, age 43 yrs. (**a**) Preoperative periapical radiograph of the right maxillary canine, with the “bone” defect presenting as a radiolucent area in both the middle of the root and the apical lesion. (**b**,**c**) Preoperative CBCT showing a radiolucent lesion on the periapical and palatal areas. This patient underwent endodontic treatment with MTA under microscopy. (**d**) Follow-up periapical radiograph 4 years later. (**e**,**f**) Follow-up CBCT 4 years later, showing filling of the root canal and healing.

**Figure 3 jpm-12-01688-f003:**
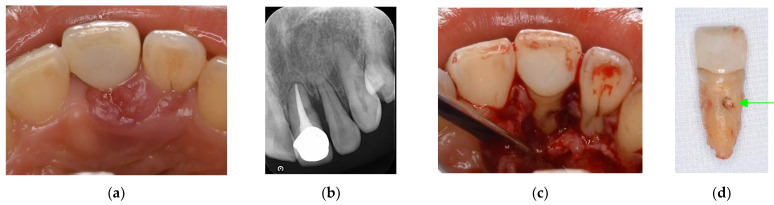
Clinical case 2. Female, age 35 yrs. (**a**) Intraoral view of the left maxillary incisor just before operation. (**b**) Preoperative periapical radiograph, with the “bone” defect presenting as a radiolucent area in both coronal and the middle of the root. (**c**) Flap retraction and debriding of the defect. (**d**) Photo of the extracted tooth, showing external root resorption (arrow).

**Figure 4 jpm-12-01688-f004:**
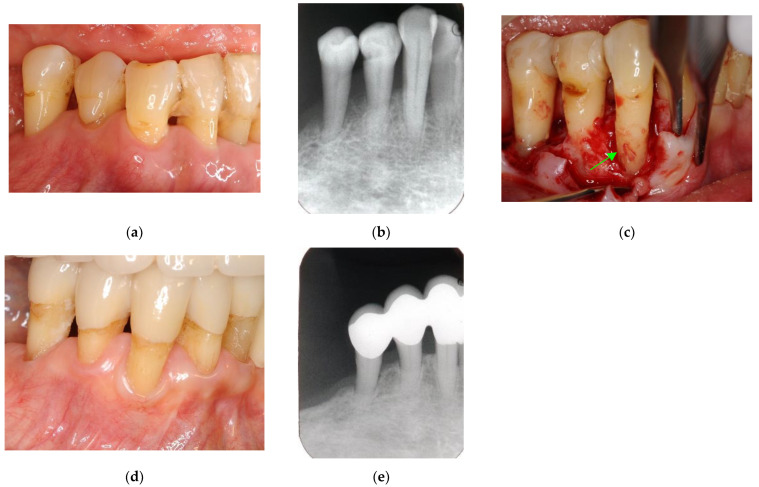
Clinical case 3. Female, age 73 yrs. (**a**) Intraoral view of the lower right canine at baseline, presenting gingival swelling and pus discharge at the mesial pocket. (**b**) Preoperative periapical radiograph showing that the “bone” defect was due to advanced periodontitis. (**c**) Flap retraction and debriding of the defect, presenting the piece of cementum (arrow). Periodontal regenerative treatment with enamel matrix derivative was performed. (**d**) Fixed prosthesis inserted 5 years later, with healthy gingiva (PD < 2 mm). (**e**) Follow-up periapical radiograph.

**Figure 5 jpm-12-01688-f005:**
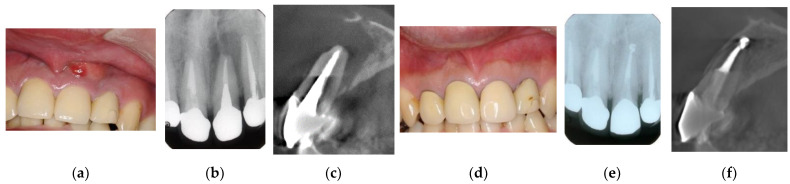
Clinical case 4. Male, age 34 yrs. (**a**) Intraoral view of the upper left incisor at baseline. (**b**) Preoperative periapical radiographs of the upper left incisor, with the “bone” defect presenting as a radiolucent area in the apical lesion. (**c**) Preoperative CBCT showing a radiolucent lesion on the labial, periapical, and palatal areas. (**d**) Intraoral view 2 years after treatment. (**e**) Follow-up periapical radiograph 2 years after treatment. (**f**) Follow-up CBCT 2 years after treatment, showing root canal filling and healing.

**Figure 6 jpm-12-01688-f006:**
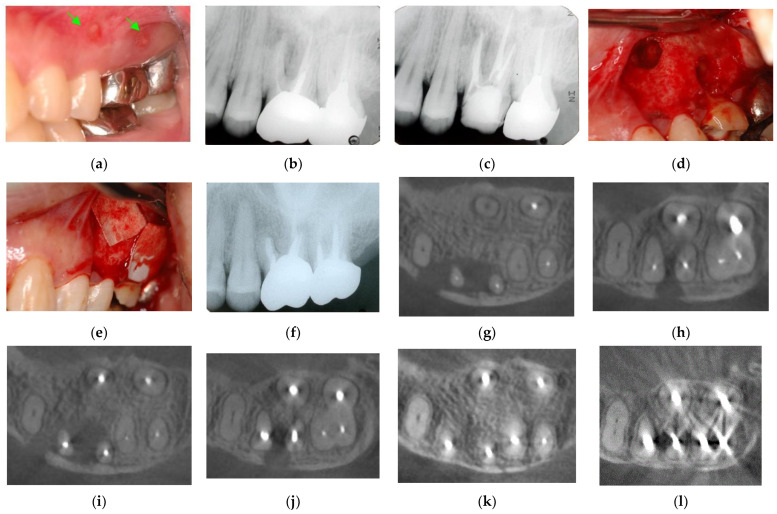
Clinical case 5. Female, age 36 yrs. (**a**) Intraoral view of the upper left first maxilla at baseline, showing two sinus tracts (arrows). (**b**) Preoperative periapical radiograph, in which the “bone” defect appeared as a radiolucent area around the MB root. (**c**) Periapical radiograph after root canal retreatment. (**d**) View after flap retraction and debriding of the defect, showing two bone defects. (**e**) GTR treatment with two resorbable membranes. (**f**) Periapical radiograph after regenerative treatment. (**g**,**h**) Preoperative CBCT showing a radiolucent lesion on the periapical and bifurcation areas. (**i**,**j**) CBCT findings after endodontic retreatment. (**k**,**l**) Follow-up CBCT 10 years after treatment, showing healing of the two bone defects.

**Figure 7 jpm-12-01688-f007:**
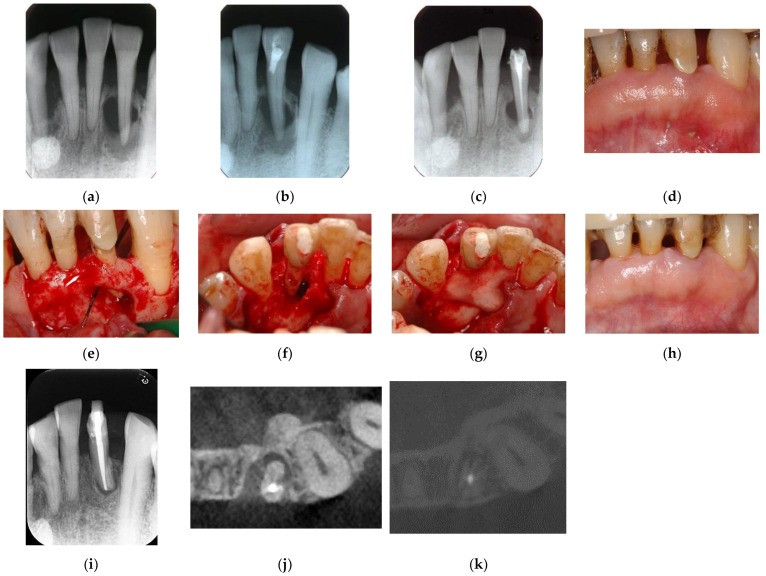
Clinical case 6. Male, age 58 yrs. (**a**) Preoperative periapical radiograph of the lower left incisor at baseline, showing the “bone” defect as a radiolucent area around the root. (**b**) Iatrogenic perforation during a previous dental procedure. (**c**) Postoperative periapical radiographs of the root canal treatment and a broken instrument. (**d**) Intraoral view 3 years after poor endodontic treatment, showing a fistula at the apical site of the tooth. (**e**,**f**) Flap retraction and debriding of the defect at both the labial and lingual sites, showing expansive bone resorption. (**g**) GTR treatment with resorbable membranes at both the labial and lingual sites. (**h**) Follow-up intraoral finding 10 years later, showing healthy gingiva. (**i**) Follow-up periapical radiograph 10 years after surgery. (**j**,**k**) Preoperative CBCT and follow-up CBCT after 10 years, showing periodontal tissue regeneration.

**Figure 8 jpm-12-01688-f008:**
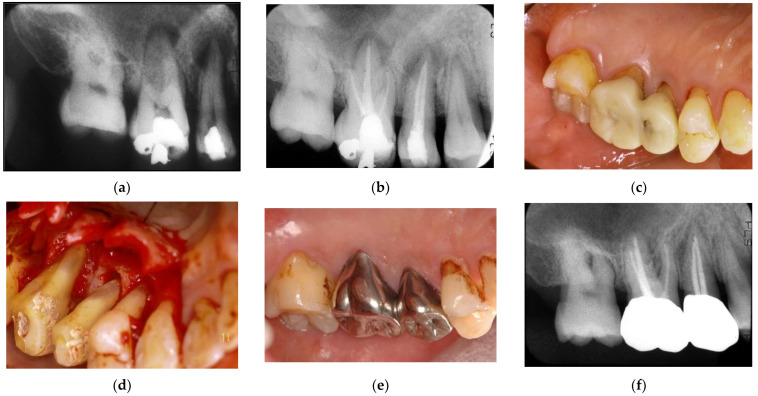
Clinical case 7. Male, age 54 yrs. (**a**,**b**) Periapical radiographs of the upper right second premolar and first molar at baseline, with the “bone” defect presenting as a radiolucent area around the root apex. In addition, deep pockets were present, and electric pulp tests were positive, suggesting retrograde pulpitis. (**b**) Periapical radiograph after root canal treatment. (**c**) Intraoral examination of the upper right molar before surgery (PD 9 mm). (**d**) View after flap retraction, debriding of the defects, and application of an enamel matrix derivative. (**e**) Final prosthesis 2 years later, presenting with healthy gingiva (PD < 3 mm). (**f**) Periapical radiograph 2 years later.

**Figure 9 jpm-12-01688-f009:**
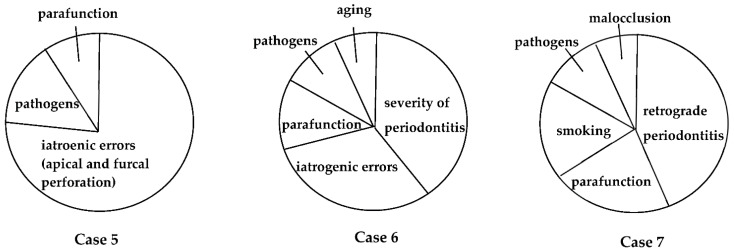
Causal/risk factors associated with EPLs in the patients described in Figure 6, Figure 7 and Figure 8. These factors may influence the progression and pathogenesis of each EPL.

**Table 1 jpm-12-01688-t001:** Systems for the classification of EPLs.

Authors	Classification Criteria
Simon et al. (1972)	Primary lesion
Guldner et al. (1985)	Etiology
Weine et al. (1989)	Treatment needs
Torabinejad and Trope (1996)	Origin of periodontal pocket
Meng (1999)	Primary lesion
Abott (2009)	Presence or absence of communication of EPL
Singh (2011)	Presence or absence of iatrogenic lesion
Al-Fouzan (2014)	Primary disease with its secondary effect
Herrera et al. (2018)	Presence or absence of root damage, periodontitis, and three grades

## Data Availability

The data presented in this study are available on request from the corresponding author.

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
