# Peer review of "Personalized Medicine Based on the Pathogenesis and Risk Assessment of Endodontic–Periodontal Lesions"

_jpm, 2022, doi:10.3390/jpm12101688_

Round 1
Reviewer 1 Report (Previous Reviewer 4)
Dear authors,
Thank you for resubmitting the paper to the journal. During the last round of reviews you were asked to remove the case presentations for the review and compose a sepparate "case series" papers. Unfortunately, you have failed to do so, please reconsider this proposal.
Kind regards
Author Response
We apologize for misleading the reviewer 1.
We had tried to show the uncertainty, controversial idea for previous classifications, possibility and future perspectives for the diagnosis and treatments of EPLs. We aimed to explain the heterogeneity of EPLs from the standpoint of personalized medicine. Therefore, this manuscript is constructed with definition, classification, other disease status, treatment modality for typical clinical cases.
When we consider personalized medicine for EPLs, we need to diagnose what kinds of EPLs and distinguish EPLs and other disease status in order to explain our basic concepts of individualized risk assessment. Clinical case 1 to 3 are other disease status from EPLs and then typical clinical findings of internal/external root resorption and cemental tears have been presented. In contrast, clinical case 4 to 7 are EPLs that showed different features as described in Figure 1 and 9. Even in EPLs, there are heterogenous risk factors among each case and then we have added clinical cases as typical models to explain our basic concept of personalized medicine for EPLs (Figure 9).
If we will separate review and clinical cases, we cannot explain our concepts described above to the readers well.
Reviewer 2 Report (Previous Reviewer 2)
The authors have addressed my comments from the previous submission. Kindly check for any grammatical or spelling errors.
A separate heading on challenges in EPL diagnosis and treatment is required to enhance the draft. Also include future perspectives
Author Response
Dear Reviewer 2
The authors have addressed my comments from the previous submission. Kindly check for any grammatical or spelling errors.
--> Thank you for your encouragements and understanding. I had obtained English editing from MPDI today.
A separate heading on challenges in EPL diagnosis and treatment is required to enhance the draft. Also include future perspectives
--> Thank you for your comments. We have added our explanation for improving our diagnosis and treatment for EPLs and future perspectives in discussion part (line 470-476).
Round 2
Reviewer 1 Report (Previous Reviewer 4)
Dear authors,
I believe your justification for including the cases is sound. Also, please consider a future case report on the topic.
Kind regards
This manuscript is a resubmission of an earlier submission. The following is a list of the peer review reports and author responses from that submission.
Round 1
Reviewer 1 Report
RE: Personalized Medicine for Endodontic-Periodontal Lesions
The topic is very interesting and has clinical applicability. However, there are some more serious errors such as author citation errors and the absence of the most recent article with the new classification of the EPL. Furthermore, the title seems to focus on personalized treatment, but treatment modalities are superficially covered in the text.
The title does not match the text. Although the authors cite about personalized medicine, most of the article deals with classification, general diagnosis, and treatment of EPL. Moreover, the treatment modalities are superficially addressed, despite the various clinical cases
In table 1, the authors described the systems for the classification of EPL, however did not include the most recent classification of EPL diseases (Caton et al. 2018)
G Caton J, Armitage G, Berglundh T, Chapple ILC, Jepsen SS, Kornman K, et al. A new classification scheme for periodontal and peri-implant diseases and conditions - Introduction and key changes from the 1999 classification. J Clin Periodontol. 2018; 45 Suppl 20:S1-S8.
In page 3, line 101, the authors say: “A more recent classification based mainly on etiology included both the cause of disease and treatment need [22].” However, the reference is from 1985. This classification it is not recent.
Some references did not correspond to the numbers.
The second paragraph of page 4 is out of context.
Author Response
The title does not match the text. Although the authors cite about personalized medicine, most of the article deals with classification, general diagnosis, and treatment of EPL. Moreover, the treatment modalities are superficially addressed, despite the various clinical cases
--> Thank you for your comment. We believe that personalized medicine for EPL could be performed with understanding the associated factors among each patient and teeth as described in Figure 4 and 9. When we manage EPL, the differential diagnosis based on the etiology is crucial and similar symptoms with different etiology such as vertical root fracture, cemental tears and iatrogenic errors such as perforation should be differentiated. In addition, treatment plan based on both endodontics and periodontics could be applied on the affected teeth, however, the difficulty of treatment and healing ability among patients are varied in fact. We decided this title after considering the limitation and uncertainty of recent dental treatment for EPL.
In table 1, the authors described the systems for the classification of EPL, however did not include the most recent classification of EPL diseases (Caton et al. 2018)
G Caton J, Armitage G, Berglundh T, Chapple ILC, Jepsen SS, Kornman K, et al. A new classification scheme for periodontal and peri-implant diseases and conditions - Introduction and key changes from the 1999 classification. J Clin Periodontol. 2018; 45 Suppl 20:S1-S8.
--> This paper (Caton et al. 2018) wrote general remarks, however, did not include the classification of EPL. Therefore, we have cited the paper by Herrera et al. (ref 30) in which new classification of EPL has been proposed as described in Table 1.
In page 3, line 101, the authors say: “A more recent classification based mainly on etiology included both the cause of disease and treatment need [22].” However, the reference is from 1985. This classification it is not recent.
--> We had rewrote the sentence as below “The classification based mainly on etiology included both the cause of disease and treatment needs had been reported [22].” according to your suggestion. (line, 99-101)
The second paragraph of page 4 is out of context.
--> We showed a real situation. We had a lot of clinical experience of EPL in which iatrogenic errors are associated with the outcome.  Especially, almost all patients who had been referral from private dental clinics to our dental hospital have iatrogenic problems including endodontic failure such as perforation, file breakage (Figure 7), inadequate dental treatment derived from clinical inertia. We rewrote the first sentence of second paragraph according to your suggestion. “Poor dental treatment especially endodontic failure could be a risk for EPL as well as peri-implantitis [33].” (line, 143-144)
Reviewer 2 Report
Major revisions are required as mentioned in the pdf

Author Response
the manuscript is too vague. Other than classification system, all other points including diagnosis, treatment, prognosis and AI are discussed in a very superficial and vague fashion. I recommend the authors revamp the entire draft to ensure facts are stated with precise stats. Based on the revised draft, I will provide further recommendation.
--> We have planned to write this review paper based on past classification, real clinical treatment based on current concept and propose possible concept for personalized medicine for EPL. EPL have wide clinical symptoms and controversy still exits for the different diagnosis and optimal treatment modality against each of cases.
is this term still used in the latest classification?
--> This paper (Papantonopoulos, et al.) has been published in 2014 and then the term of aggressive periodontitis” had been used in general. In addition, the Stage and Grade system had been proposed in the World Workshop for Classification of Periodontal and Peri-implant Diseases and Conditions at 2017 and this is just a baby situation and may be modified in the future with new information. Therefore, we use the both term in Japan at present.
vertical root fracture?
--> Yes, we used “VRF” instead of “vertical root fracture”. (line 130-131)
throughout the manuscript provide a estimate % instead of using generic terms like common frequent or lower
--> Thank you for your useful comments. We rewrote as described below according to your comments. “The iatrogenic errors, such as ledges and root perforations has been reported and the frequency of errors was significantly greater in molars than in anterior teeth [39, 40]. The success rates are the lowest in molar (26%) compared with premolars (49%) and anterior teeth (52%) because of the complex anatomy of the root canal system in molars and the difficulty of endodontic treatment by dentists not familiar with the proper procedures [41, 42]. (line, 195-200)
define/describe it in the brackets
--> Thank you for your useful comments. We rewrote as described below according to your comments “nonsurgical endodontic treatment by using an operating microscope” (line, 208)
info provided under this heading is too vague. A more comprehensive and specific discussion on the risk assessment for EPL is required
--> We agree with the reviewer’ comment, however, there are few information of risk assessment of EPL at present, therefore, we tried to propose our concepts based on current knowledge and our clinical experience as a specialist of endodontist and periodontist.
elaborate on these charts to provide some context
--> Thank you for your comments. However, we do not have enough space to explain this chart in detail and then added the citation for readers.
is this the correct term to be used ?
--> We think this term “hopeless” is clinically acceptable in general. (line, 293)
authors have to provide accurate % stats.
--> Thank you for your comments. However, Kim et al. (ref. 51) did not mention the success rate of EPL, although they wrote the 96.8% healing success after 1 yr of periapical lesions without EPL.
u need to elaborate
-->Thank you for your comments. We expect the AI-assisted diagnosis for many diseases performed in medicine could be applied in our dental medicine, however, we do not have enough information at present as you know.
expand on this. provide more info
--> Unfortunately, there are few information at present, however, we expect future advance of AI-associated devices.
based on what data was this statement made?
--> Thank you for your comments. We have cited two papers (ref. 39, 41) and rewrote this sentence.
Reviewer 3 Report
Comments:
The present manuscript concerns a relevant topic, the endodontic-periodontal lesions, from an interesting point of view, which is personalized medicine.
The study type is not completely clear: the narrative review is mixed with a case series. Therefore, I would strongly suggest to divide these two parts in the text, with the review first followed by the cases presentation (including Figures), discussed with the argumentations of the review, providing definitively clinical recommendations.
Manuscript should be reorganized.
Editing for English language is needed.
Reviewer’s concerns:
· Concerning periodontitis, which “has a multifactorial nature, with patients and teeth differing in suceptibility and resistance to periodontitis depending on inidividual causal and risk factors [7]” (lines 40-41), please discuss the role of general health conditions, medicaments, etc. see
Obesity and periodontal disease: a narrative review on current evidence and putative molecular links DOI: 10.2174/1874210601913010526 and
Periodontal and peri-implant diseases and systemically administered statins: a systematic review. Dentistry Journal. 2021, 9(9), 100 DOI: 10.3390/dj9090100.
“world workshop for classification of periodontal and peri-implant 133 diseases and conditions” should be capitalized.
· Please, remove the period “Poor dental treatment may be a major risk for EPL as well as peri-implantitis [33]. 143 The prevalence of endodontically treated teeth was higher in Japan than in Europe and 144 the USA [34] and iatrogenic errors have been associated with refractory periapical 145 periodontitis in general practices in Japan [35]. Both endodontic and periodontal 146 treatment may injure treated teeth, emphasizing the need for continuous training of 147 dentists.”.
· Please, rephrase the sentence “Technical and surgical skills vary widely among dentists, making them difficult to 374 determine objectively.”.
· Please, correct “collabolations” (line 403).
· The following paragraphs should be moved at the end of the cases presentation “Personalized medicine aims to individualize and optimize treatment and care based 477 on each person’s unique characteristics, including genetic, environmental and clinical 478 profiles [60]. Evidence based medicine is based on a similar idea of optimizing 479 individualized medicine, by integrating each individual clinician’s expertise with the best 480 scientific evidence available [61]. Clinical experience, diagnostic ability and excellent 481 surgical skill are crucial for treatment of EPL, especially in teeth deemed truly hopeless. 482 The primary goal of EPL treatment must be to remove infection. In addition, optimal 483 outcomes require the management of causal/risk factors in individual patients (Figure 9). 484 The ability of patients to heal also varies, and systematic conditions such as diabetes 485 mellitus may reduce their ability to heal and worsen outcomes of endodontic treatment 486 [62, 63]. Additional basic and clinical research are required to assess the associations of 487 treatment outcomes with the healing ability of patients.”.
Author Response
Comments:
The present manuscript concerns a relevant topic, the endodontic-periodontal lesions, from an interesting point of view, which is personalized medicine.
The study type is not completely clear: the narrative review is mixed with a case series. Therefore, I would strongly suggest to divide these two parts in the text, with the review first followed by the cases presentation (including Figures), discussed with the argumentations of the review, providing definitively clinical recommendations.
Manuscript should be reorganized.
--> Thank you for your advice. I agree with your comments, however, we try to describe and highlight the complexity and uncertainty of diagnosis and decide optimal treatment against each cases at present. Therefore, we showed historical changes of classification of EPL and similar diseases to EPL and recommend personalized medicine for each EPL as an optimal treatment strategy with real clinical cases that had been treated by myself.
Editing for English language is needed.
--> We had already asked English editing.
Reviewer’s concerns:
- Concerning periodontitis, which “has a multifactorial nature, with patients and teeth differing in suceptibility and resistance to periodontitis depending on inidividual causal and risk factors [7]” (lines 40-41), please discuss the role of general health conditions, medicaments, etc. see
Obesity and periodontal disease: a narrative review on current evidence and putative molecular links DOI: 10.2174/1874210601913010526 and
Periodontal and peri-implant diseases and systemically administered statins: a systematic review. Dentistry Journal. 2021, 9(9), 100 DOI: 10.3390/dj9090100.
--> Thank you for your useful advice and information. I had read these papers. Although these papers are interesting, we do not have enough space to cite and discuss the context of papers unfortunately.
“world workshop for classification of periodontal and peri-implant diseases and conditions” should be capitalized.
--> We had capitalized according to your suggestion. (line, 133-134)
- Please, remove the period “Poor dental treatment may be a major risk for EPL as well as peri-implantitis [33]. The prevalence of endodontically treated teeth was higher in Japan than in Europe and the USA [34] and iatrogenic errors have been associated with refractory periapical periodontitis in general practices in Japan [35]. Both endodontic and periodontal treatment may injure treated teeth, emphasizing the need for continuous training of dentists.”.
--> We had convinced iatrogenic errors, especially endodontic failure, could be a risk for EPL as described in Figure 7. Therefore, we rewrote the first sentence of second paragraph according to your suggestion. “Poor dental treatment especially endodontic failure could be a risk for EPL as well as peri-implantitis [33].” (line, 143-144)
- Please, rephrase the sentence “Technical and surgical skills vary widely among dentists, making them difficult to determine objectively.”
--> Thank you for your comments. We rewrote these sentences according to your suggestion. “The uncertainty of diagnosis and dental treatment make decision making difficult. Therefore, providing this information to patients may better enable them to select dentists suitable to treat their dental problems. (line, 374-376)
(line 403).
- The following paragraphs should be moved at the end of the cases presentation “Personalized medicine aims to individualize and optimize treatment and care based on each person’s unique characteristics, including genetic, environmental and clinical profiles [60]. Evidence based medicine is based on a similar idea of optimizing individualized medicine, by integrating each individual clinician’s expertise with the best scientific evidence available [61]. Clinical experience, diagnostic ability and excellent surgical skill are crucial for treatment of EPL, especially in teeth deemed truly hopeless. The primary goal of EPL treatment must be to remove infection. In addition, optimal outcomes require the management of causal/risk factors in individual patients (Figure 9). The ability of patients to heal also varies, and systematic conditions such as diabetes mellitus may reduce their ability to heal and worsen outcomes of endodontic treatment [62, 63]. Additional basic and clinical research are required to assess the associations of treatment outcomes with the healing ability of patients.”.
--> I understood your comment, however, this paragraph has been composed from established concepts, EBM and our clinical experience. Therefore, this means our conclusive opinion and then the position of the end of discussion is optimal.
Reviewer 4 Report
Dear authors,
Thank you for submitting your valuable work to the journal. The topic of your review is intersting and can add good information to everyday dental practice, where the prevalence of ELPs is significant. However there are some comments I would make in order to improve the paper's scientific accuracy:
- please rephrase and improve clarity of review's objectives at the end of the Introduction
- please include a Material and Method section, briefly describing the methodology used for this review. PRISMA guidelines should be included and followed
- it is not clear if Clinical Cases are original, although they are good for reflecting the information, they shouldn't be included in the Review
- please check formating and English language throught the manuscript
We look forward to receiving the revised version of your manuscript!
Kind regards
Author Response
- please rephrase and improve clarity of review's objectives at the end of the Introduction
--> Thank you for your useful comment. We had rewritten the last paragraph of introduction “The objectives of present review highlight the potential causes and risk factors associated with EPL and the uncertainties involved in classifying each EPL because of its multifactorial nature. Moreover, this review proposes practical methods of determining treatment and the advantages of precise and personalized medicine for each EPL based on risk assessments in individual patients and an interdisciplinary approach. ”
- please include a Material and Method section, briefly describing the methodology used for this review. PRISMA guidelines should be included and followed
--> Thank you for your useful comment. However, in this review paper, we did not perform systematic review or meta-analysis and then PRISMA guidelines are not essential in this paper.
- it is not clear if Clinical Cases are original, although they are good for reflecting the information, they shouldn't be included in the Review
--> Thank you for your useful comment. The clinical cases presented in this review had been performed by myself. We had included several clinical cases for helping leaders to understand our diagnostic and therapeutic philosophies based on our clinical experience.
- please check formating and English language throught the manuscript
--> We had already done it.
Round 2
Reviewer 1 Report
I suggest the reject of article.
Reviewer 2 Report
The revisions are to my satisfaction.
Reviewer 3 Report
Dear Authors,
I still find that the type of the manuscript is not defined. The methology is confused, although the topic is relevant and the cases presented are very interesting.
I would suggest again to separate the narrative review from the case series and apply the currently accepted classification.
Reviewer 4 Report
The authors have responded to my comments accordingly. The paper has improved significantly. Thank you!